# Historical cartographic and topo-bathymetric database on the French Rhône River (17th–20th centuries)

Fanny Arnaud[1], Lalandy Sehen Chanu[1], Jules Grillot[1], Jérémie Riquier[2], Hervé Piégay[1], Dad Roux-Michollet[3], Georges Carrel[4], Jean-Michel Olivier[5]

[1]Université de Lyon, CNRS UMR 5600 EVS, École Normale Supérieure de Lyon, Lyon, F-69342, France
[2]Université de Lyon, CNRS UMR 5600 EVS, Université Jean Monnet Saint-Etienne, Saint-Etienne, F-42023, France
[3]Syndicat du Haut-Rhône, F-73170, Yenne, France
[4]INRAE, RECOVER, F-13182, Aix-en-Provence, France
[5]Université de Lyon, Université Claude Bernard Lyon 1, CNRS, ENTPE, UMR5023 LEHNA, F-69622 Villeurbanne Cedex, France

*Correspondence to*: Fanny Arnaud (fanny.arnaud@ens-lyon.fr)

## Abstract

Space and time analyses of channel changes, especially within large rivers subject to high levels of human impact, are critical to address multiple questions about rivers in the Anthropocene era. The reconstruction of long-term (>150 year) evolutionary trajectories permits an understanding of how natural and anthropogenic factors impact hydromorphological and ecological processes in rivers, helps with the design of sustainable management and restoration options, and may also help in the assessment of future changes. However, the reconstruction of channel changes can be challenging: historical data are often scattered across many archives, and the quantity and accuracy of information generally decreases as one goes back in time. This data article provides a historical database of 350 cartographic and topo-bathymetric resources on the French Rhône River (530 km in length) compiled from the 17th to mid-20th century, with a temporal focus prior to extensive river training (1860s). The data were collected in 14 national, regional, and departmental archive services. A table describes the properties of each archived data item and its associated iconographic files. Some of the historical maps are available in a georeferenced format. A GIS layer enables one-click identification of all archive data available for a given reach of the French Rhône River. This database provides substantial new material for deeper analyses of channel changes over a longer time period and at a finer time step compared with previously available data. The database has several potential applications in geomorphology, retrospective hydraulic modelling, historical ecology, and river restoration, as well as permitting comparisons with other multi-impacted rivers worldwide. The dataset is available at doi 10.1594/PANGAEA.922437 (Arnaud et al., 2020a). Iconographic extracts of the 350 archived items are available at http://photo.driihm.fr/index.php/category/52.

**1 Introduction**

Characterization of the evolutionary trajectories of the world's rivers and floodplains is crucial for assessment of space-time changes in biogeomorphic processes, the quantification of anthropogenic impacts, and the planning of sustainable management based on conservation and restoration options (Gabrowski et al., 2014; Mould and Fryirs, 2018). The analysis of long-term (>150 year) channel changes provides a holistic understanding of the trajectory of fluvial hydrosystems and their drivers, which helps in defining process-based restoration objectives, rather than determining static reference states (Dufour and Piégay, 2009;

Eschbach et al., 2018). It is necessary to extend the timescale of past trajectories to characterize possible shifts in baseline conditions related to climate variability and different degrees of human pressures to help in the assessment of future river changes in the Anthropocene (Pont et al., 2015; Slowik, 2015; Piégay et al., 2020).

The reconstruction of river trajectories requires substantial historical datasets. Series of maps, aerial images, topographic, and bathymetric surveys are generally selected at regular time steps, if available, or before and after major events (i.e. floods, river

engineering phases) to assess changes in the river hydromorphological conditions. This set of archival materials is used to extract channel features such as active channel width, bar-island areas (Hohensinner et al., 2004; Michalkova et al., 2011; Scorpio et al., 2015; Lestel et al., 2018; Hohensinner et al., 2021), bed degradation/floodplain aggradation rates (Steiger et al., 2001; Downs et al., 2013; Arnaud et al., 2015), riparian vegetation patterns (Cadol et al., 2011; Belletti et al., 2013; Kui et al., 2017; Safran et al., 2017), and landscape unit characteristics (Dufour et al., 2015; Solins et al., 2018; Piégay et al., 2020).

Following Roux et al. (1989) and Habersack and Piégay (2008), comparative analyses of river changes, notably in Europe, have become a critical issue for addressing a series of questions of interest to geographers, historians, engineers, ecologists, geologists, and geochemists. For example, historical maps can help to better contextualize sediment core samples and geophysical surveys when investigating floodplain evolution (Salvador and Berger, 2014) and delta evolution during the Holocene (Vella et al., 2005). Historical cartographic and topo-bathymetric datasets are also of interest to hydrologists

analyzing flood frequency based on historical geometries (Billy et al., 2018). They are also key resources in historical ecology, allowing the formulation of hypothesis on fish habitat-species relationships since the industrial revolution, especially given the lack of old quantitative data in terms of species number, abundance, and distributions (Carrel, 2002; Frimpong et al., 2016; Belliard et al., 2018).

A number of historical analyses have been conducted on the Rhône River in Western Europe. Whereas the Rhône has been a

braided and anabranching river in multiples reaches along its 530 km-long French course since the 17th century, it has been deeply regulated for 150 years, and is now targeted by large-scale restoration measures aimed at improving morpho-ecological processes (Riquier and Cottet, 2019). The mid-19th century period is usually considered the benchmark for fluvial patterns before extensive river training. A detailed map series was plotted by the *Ponts et Chaussées* (Bridges and Roads) public administration, in the 1860s, at the scale of 1:10 000. It provides data on (i) the morphological pattern of the Rhône and its

former branches, (ii) the type of land cover on the floodplain, and (iii) topo-bathymetric information (Bravard, 2010). Most diachronic studies have used this 1860's map series as a starting point for detecting geomorphic changes, by making

comparisons with later maps surveyed at the end of the 19th century and mid-20th century, and aerial images taken since the 1930s (Bravard et al., 1986; Bravard, 2010; Provansal et al., 2014; Parrot, 2015; Dépret et al., 2017; Räpple, 2018; Thorel et al., 2018; Tena et al., 2020). Therefore, the post-1860 planform adjustments are fairly well understood and are combined with vertical adjustments measured on reaches where topo-bathymetric information has been available since the late 19th century. However, little is known about the morphodynamics of the Rhône in the early 19th century and times before this.

This paper presents a historical database of 350 cartographic and topo-bathymetric resources covering the French Rhône River, collected from 14 national, regional, and departmental archive services. The timescale extends from the 17th to the mid-20th centuries, with a pre-1860 focus for cartographic data and a pre-1900 focus for topo-bathymetric data. The aim of this paper is to provide substantial new material to extend and detail historical trajectories of the Rhône River for a wide range of research applications including historical hydrology and historical biogeomorphology, within flood management and river rehabilitation contexts. The compiled database will enable further comparisons with other large braided/anabranching multi-impacted rivers worldwide.

## 2 Study area

The Rhône is an 812-km long river connecting the French-Swiss Alps with the Mediterranean Sea. Originating at the Furka Glacier, the alpine Rhône drains the Valais region up to Lake Geneva. The lower 534 km of the river course are in France. As a whole, the catchment size is 98 500 km², with 90 500 km² being in France (Olivier et al., in press) (Fig. 1). The river delimits or crosses 11 French administrative departments. It drains areas with a wide range of geological, altitudinal, hydro-climatic and land-use conditions, which results in a heterogeneous channel reach characterized by alternating fixed and mobile channel beds, incised valleys, and wide floodplains, and distinct sediment and flow inputs from tributaries (Bravard, 2010; Riquier and Cottet, 2019). The mean bed slopes are 0.9 % on the Swiss river course, 0.1 % on the upper Rhône section (Lake Geneva to Lyon-Saône confluence), 0.05 % on the middle Rhône section (Lyon to Valence-Isère confluence), 0.06 % on the lower Rhône section (Valence to Arles-Gard confluence), and 0.004 % at the delta (Parrot, 2015).

The mean annual discharge is 1720 m$^3$ s$^{-1}$ at the most downstream gauging station (Beaucaire), which makes the Rhône the most important freshwater input into the northwestern Mediterranean Sea (Van Den Broeck and Moutin, 2002). The 100-year flood is 10 300 m$^3$ s$^{-1}$. Highest floods since the beginning of daily hydrological measurements, from 1816 at Beaucaire, have occurred in October 1840 (13 000 m$^3$ s$^{-1}$) and May 1856 (12 500 m$^3$ s$^{-1}$). These floods are considered as a "reference" for the contemporary period. The mid-19th century corresponds to the end of the Little Ice Age, a period during which flood activity was high: the 1701–1710 period has especially exhibited the highest flood frequency of the Rhône history since the 14th century (22 months per decade at Arles against 13 months per decade for the 1851–1860 period; Pichard, 1995; Arnaud-Fassetta, 2003). The floods of November 1694 and December 1755 are the highest of their centuries (Pichard and Roucaute, 2014). The flow regime of the Swiss alpine Rhône is snowmelt-dominated, with low flows from November to April and high flows in late spring and summer. The French upper Rhône and the middle Rhône sections are influenced by a wide diversity of tributaries

showing several flow patterns, like ice and snowmelt (Isère, Arve rivers) and rainfall (Saône, Ain rivers). On the lower Rhône, the major tributaries (Drôme, Ardèche, and Durance rivers) have a Mediterranean regime with high discharges mostly during spring and autumn and frequent flash floods (Bravard, 2010).River training was conducted along both the Swiss and French parts of the Rhône. In France, human interventions were developed during three main periods (Thorel et al., 2018; Tena et al., 2020): (1) in the second half of the 19th century, with the construction of longitudinal 'unsubmersible' dykes to locally protect the population from flooding; (2) from the late 19th to the early 20th century with the systematic construction of groyne fields connected to submersible dykes (infrastructures named 'Casiers Girardon' from the engineer who designed them), created almost continuously from Lyon to Arles, and aimed at concentrating flows into a narrow single-bed channel to improve navigation; and (3) from the 1920s to the 1980s, with the construction of 19 hydro-electric dams and by-passing schemes between Lake Geneva and the Mediterranean Sea by the National Company of the Rhône River (CNR).

Channel correction and flow regulation induced major hydromorphological modifications such as incision and armouring, disconnection of secondary channels, floodplain terrestrialization, and subsequent loss of ecological functionalities (Roux et al., 1989; Amoros and Bornette, 2002; Parrot, 2015; Dépret et al., 2017; Tena et al., 2020). A large-scale restoration plan for the French Rhône has been conducted since 1998 to improve ecological conditions in the main channel and floodplain. This plan is based on the combination of two main measures: (i) increasing the minimum flow in by-passed reaches, and (ii) reconnecting abandoned channels (Lamouroux et al., 2015). A second phase of restoration was launched in 2009, and currently consists of enhancing sediment transport processes into by-passed and main channel sections. The measures include riprap removal to reinitiate bank erosion, and reintroduction of gravels dredged from reservoirs and navigable channels or excavated from restored side channels. Monitoring programs are co-constructed between researchers and practitioners, to assess restoration success and provide insights to improve restoration strategies (Lamouroux et al., 2015; Thorel et al., 2018).

## 3 Material and methods

### 3.1 Data collection

This study started with an inventory of historical resources compiled in previous long-term research: the RhônEco program (2001–present), Rhône Sediment Observatory (2009–present), and Rhône Valley Human-Environment Observatory (2011–present). We found one long profile and 15 map series collected between 2009 and 2015 from the CNR, the Regional Directorate for Environment, Development and Housing (DREAL), the National Institute of Geographic and forest information (IGN), and the Departmental Archives of Savoie (AD73), Drôme (AD26), Ardèche (AD07), Isère (AD38), Vaucluse (AD84), and the University of Grenoble (Univ. Grenoble). Most of the maps were scanned and integrated into a GIS interface. Their spatial coverage extends from about 10 km to the entire 530-km length of the French river course. The oldest map series is the Sardinian Maps, consisting of twenty communal maps of the left bank of the upper Rhône surveyed at the scale of 1:2372 (the original scale is in 'trabucs') by the Kingdom of Piedmont-Sardinia in 1728–1738. The most recent map series are the CNR

pre-work plans, consisting of 128 sheets surveyed between 1946 and 1983 at the scale of 1:10 000 to plan 19 damming and by-passing schemes on the Rhône River from Injoux-Génissiat to Arles.

Between 2017 and 2020, we searched archive services that had been only poorly explored previously: the National Archives of Pierrefitte-sur-Seine (AN), Bibliothèque nationale de France (BnF), and the Departmental Archives of Rhône (AD69), Ain (AD01), and Gard (AD30). The AD Rhône holds the main collection of the *Ponts et Chaussées* administration/Rhône Navigation Service. This consists of numerous maps, plans, and profiles regarding public works and authorizations in the public fluvial domain over the entire river course. We consulted the entire sub-series *S/Public Works (1784-1988)* and *3959 W 1-2030/Rhône-Saône Navigation Service (1807-1994)*. Other relevant references were identified in the AD Rhône numeric catalogue by entering the keywords 'Rhône River' and 'map', notably throughout the *Ancien Régime* (16th–late 18th century). In total, 113 archive boxes were consulted. This research was completed at the National Archives, which also hold the *Ponts et Chaussées* collection. In terms of spatial extent and survey period, the online catalogue contained less information than the AD Rhône one. However, it was possible to identify adequate references with the keyword 'Rhône River' in the sub-series *F/14/Public Works*. Not all resources were described online; thus, the paper registers of the sub-series *F/14/Public Works* and *F/10/Agriculture* were also consulted. The other archive services were similarly searched by keywords, in both numeric catalogues and paper registers. A total of 211 archive boxes were consulted, with each box containing a bundle of documents for either a specific sector at different periods, or a set of surrounding sectors at a given period (Fig. 2). The documents included letters, newspaper extracts, engineer reports, survey tables, cost estimates for further river management, and iconographic resources generally produced to survey flood damage or to plan new infrastructure. We systematically took pictures of iconographic resources, which included maps (Fig. 3.a, b), cross-sections, long profile series (Fig. 3.b), and topo-bathymetric information drawn on maps (Fig. 3.c).

The series *CP* (Maps and Plans Service) of the National Archives was given special attention. This series contains older, larger and/or fragile maps, field surveys, and engineering structure iconographies, and requires specific authorization for consultation (Fig. 3.b), with the resources having to be consulted one by one. Because of time constraints, we focused on maps covering long river reaches rather than detailed maps of ports and cities, and focused on reaches involved in present river restoration projects, notably those of Péage-de-Roussillon and Donzère-Mondragon. We selected 33 resources amongst the 53 available for consultation in the sub-series *CP/F/14/Streams and Rivers* (6.5 p.). The sub-series *CP/F/14/Canals* had 12 pages dealing with a project for a continuous lateral canal from Lyon to the Mediterranean Sea, which was finally dropped, and 3 pages dealing with the canal from Arles to Port-de-Bouc at the Rhône delta. We selected 39 resources amongst the 113 available for consultation. The box references consulted in all the archive services are listed in the dataset at doi:10.1594/PANGAEA.922437 (Arnaud et al., 2020a).

All collected data were photographed on a table or on a magnetic board in the case of multiple-sheet and large documents, and were saved in .jpg format. The most interesting data were scanned at high resolution (.JPEG and .TIFF format, up to 3 GB each).

## 3.2 Database structure

Table 1 describes the properties of the 350 archived data items available in the Arnaud et al. (2020a) dataset. This table has 18 columns that include the following: (1) the type of data: i.e. map, cross-section, long profile, table, topo-bathymetry drawn on the map, (2) the source (archive service) and reference (box number), (3) the original title (in French) and a translated English title, (4) the scale, author, and survey period, (5) the communes covered by the upstream-downstream Kilometric Points (KPs; km 0 is in Lyon, distances upstream from this reference point are negative and distances downstream are positive; the KPs were identified in ArcMap 10.5 using the GIS layer of riverine communes and active channels in 1860, to help in locating the maps with secondary channels). (6) comments on the quality and content of the data, (7) the number of images and their type (on a table or a magnetic board, HD scan, scanned and georeferenced), and (8) the storage file and sub-file.

The iconographic database available in Arnaud et al. (2020a) is 12.7 GB. It contains 3437 images stored in 32 files and 342 sub-files. Thirteen files are named after the engineer who did the surveys. For example, the 'Kleitz (1853-1864)' and 'Bouvier (1785-1848)' files contain 63 and 24 sub-files, respectively. These files describe maps of different reaches, sometimes associated with topo-bathymetry data. The '*Ponts et Chaussées* (1807-1858)' file contains 37 maps surveyed by authors who are unknown or whose names are unreadable. The database also contains 14 files corresponding to a single engineer survey, generally at scales of 1:10 000 or 1:20 000 along a long river course (from 60 to 530 km), e.g. the map by Josserand surveyed in 1845 between Lyon and Donzère (170 km long), or the long profile by the *Ponts et Chaussées* administration surveyed in 1869 from Seyssel to Lyon (151 km long; Fig. 3.c). Three files consist of maps of the three main reaches of the Rhône: the French upper Rhône (16 maps from 1730–1875), the middle-lower Rhône (13 maps from 1642–1857), and the delta (16 maps from the 18th–19th century). Finally, the file 'Topo-Bathy (1799-1882)' contains 19 cross-section and long profile series surveyed by different authors.

The 350 archive resources include 330 maps and 93 topo-bathymetric data items (Fig. 4.a). About forty other documents were also collected, corresponding to plans (<1:1000), information on the Swiss Rhône, and books from the late 19th and early 20th centuries on fish populations (Vingtrinier, 1882; Locard, 1901; Marchis, 1929). Maps and topo-bathymetric data mainly cover the 1840–1855 period (Fig. 4.b), which corresponds to the beginning of river training after the 100-year flood in October 1840. This flood was responsible for significant damage, and led to the establishment of the Rhône Special Service within the *Ponts et Chaussées* administration in 1840 (Bravard, 2010). The 1856–1870 period was also well surveyed, following the other 100-year flood in May 1856. Nineteen percent of the database covers the 1640–1839 period. The oldest resource is a series of 6 maps found in the Chaponnay private archive of AD Rhône, surveyed in 1642–1660, and depicting the 'brotteaux' (grazing areas on recent alluvial deposits) between Pierre-Bénite and Irigny fluvial ports to the south of Lyon (Fig. 5). This map series represents the Rhône River in an artistic manner; however, the distances between villages were checked and were found to be realistic.

Hydrological information like spatial extent of large floods and the associated water-level at gauging stations is provided on most maps. On the opposite, the discharge of the aquatic channel is rarely given, and we have to assume that the river is

represented at its the mean annual discharge, as usual in cartography. Only few maps, like the 'Topographical map of the Rhône River' surveyed in 1857-1876 by the *Ponts et Chaussées* administration indicate in the chart legend that the channel was mapped at the mean annual discharge. Finally, water-level information is systematically provided on topo-bathymetric data, i.e. the water level on the day of the survey, the low-flow level and/or the high flood levels (Fig. 6).

Fig. 7 provides a synoptic view of all cartographic data with corresponding spatial scales along the Rhône River. Most items were surveyed between the scales of 1:1000 and 1:20 000, with 1:10 000 being mainly used. This latter scale is suitable for quantitative analysis of morpho-ecological units. Similarly, Fig. 8 depicts the longitudinal coverage of topo-bathymetric data.

### 3.1 GIS integration

The longitudinal coverage of each of the 350 data items was positioned along the Rhône River centreline in ArcMap 10.5, according to the upstream and downstream KPs. The individual coverages were then merged into one polyline layer available in the Arnaud et al. (2020a) dataset. This layer provides efficient identification, at a single click, of archive resources available for a given reach. For example, dragging a selection box over the Chautagne by-passed reach (KP −146 to −137) displays 24 data items on this reach (Fig. 9).

The Arnaud et al. (2020a) dataset provides 14 scanned map series, in both raw (.JPEG or .TIFF) and georeferenced (.GeoTIFF) formats (Table 2). They are among the most spatially extensive and accurate maps in terms of the features represented. Half of these maps have been georeferenced in previous works of the Rhône observatories, using a first order polynomial transformation (between 3 and 15 Ground Control Points [GCPs] per image, RMS error up to 50 m). In this study, we tested the spline transformation (named Thin Plate Spline transformation in QGIS) based on recommendations of Lestel et al. (2018) for old maps. The spline transformation is a true rubber-sheeting method optimized for local but not global accuracy. It transforms the source GCPs exactly to the target GCPs, with minimal error around the points. The spline transformation is therefore useful when the number of GCPs is important (www.esri.com). The spline transformation was tested on map series of 1760 and 1831 by adding GCPs, and was compared with the polynomial transformations (Table 2). The positioning on the present orthophotograph was significantly improved, and the spline transformation was therefore applied to the newly collected maps. Raw and georeferenced images are usable in both ArcGIS and QGIS softwares. Among the 316 other maps of the database, most can be georeferenced by using riverine roads, buildings and dykes as GCPs. When available, land register can also be used. Though, maps surveyed at a too large scale (1:1 000, 1:2 000), or those representing a single bank of the Rhône River, do not have enough GCPs to be georeferenced.

# 4 Data use and application

The type of archive data, either manually photographed or scanned, and in some cases georeferenced, allows the user a qualitative or quantitative analysis, including illustration of the fluvial landscape of a large river at the beginning of the 19th century, evaluation of the main transformations following river regulation, and diachronic GIS extraction of metrics such as channel widths, secondary channel numbers, and gravel bar areas. The scanning and georeferencing of the sources are not exhaustive. The archive references are accurately described in the table of data properties to facilitate easy consultation and 225   scanning of data of interest.

  The dataset depicts heterogeneous information, which depends on the survey date and map objectives. A critical analysis should therefore be carried out before any use of the dataset on a given river reach. As an example, Fig. 10 compiles maps surveyed in the upstream section of the Donzère-Mondragon (DZM) reach, which is currently involved in side channel reconnections, riprap removal, and gravel augmentation. The 5-km-long section (KP 171 to 176) embraces the eastern Bayard 230   secondary channel, which was the main course of the river in the late 18th century. Among the 26 maps available prior to 1860, nine have been selected at scales of 1:5 000–1:10 000. Maps from 1810 to 1842 do not specify the nature of islands. Partial information is given on the 1843 map, with gravel bars drawn in light orange. Symbols are then used on the 1846 map to differentiate gravel bars and vegetated areas. These symbols are also used on the 1852 and 1856 maps: their base map is the 1846 edition superimposed with the new channel pattern and engineering infrastructure, which is a common practice in 235   engineering cartography (Lestel et al., 2018). The 1856 map was edited to report damage to 'unsubmersible' dykes after the flood of May 1856. New islands and gravel bar areas formed after the flood were drawn in light orange. Note that the titles of the maps are equivocal: 71 maps of the database have the word 'dyke' in their title and were surveyed to report damages after large floods, rebuild dykes or design new ones. The challenge of map-based historical studies is to use individual sources over time, with sometimes little information on the context in which the map was created, and therefore on its certainty level. The 240   comparison of multiple maps available at close dates (some of them surveyed in the same year by different engineers) allows inconsistencies between individual sources to be resolved while revealing persistent landscape features and patterns (Safran et al., 2017). For example, the coloured map of 1842 from Switzerland to the Mediterranean Sea surveyed by the engineer Arnaud is fairly aesthetic; however, a comparison with other maps highlights the lack of accuracy in the depiction of geomorphic features, notably on the downstream part of the Bayard channel, as well as the non-exhaustive mapping of roads and villages, 245   which limits the number of ground control points available for georeferencing. In addition, when analyzing long-term vegetation changes, it is important to carefully examine vegetated areas representation (when available) because vegetation could be represented in many different ways, either by colors, symbols or annotations (Fig. 10 and 11). In most cases, only vegetated islands and the riparian vegetation close to the channel were mapped (Fig 11a and 11b). The depiction of vegetation in the entire floodplain was less frequent (Fig. 11c), especially regarding vegetation types (grassland, shrubs, forest, 250   agricultural areas; Fig 11.d). Note that the more recent maps did not have systematically more information than the older ones.

Keeping in mind the preliminary examination of the data accuracy, the large number of historical data items collected in this study should enable deeper analyses of channel shifting, island dynamics, and landscape changes over a longer time period and at a finer time step than those performed in previous studies, and also on many reaches of the Rhône River and tributary confluences for which no historical maps had been collected until now. The archives prior to 1860 provide us with historical benchmarks for process-based river restoration, and permit comparisons with other rivers with available archives. A potential application could be the detailing of past trajectories of floodplain units, such as that performed by Tena et al. (2020) on three by-passed reaches of the middle and lower Rhône (Fig. 12). Similarly, Dépret et al. (2017) combined historical maps and aerial images (1860-2009) to date a series of channel disconnections and study terrestrialization processes. They showed that terrestrialization of naturally cut-off channels was much quicker than that observed on artificially cut-off channels, probably because of much higher bedload transport, which promoted active sediment infilling. The historical dataset is also valuable to explore longer-term ecological succession, such as initiated by Bravard et al. (1986) and Roux et al. (1989) on the upper Rhône section. During the 18th and 19th centuries, the backswamps and the gravel bars of the braided belt were devoted to extensive grazing, and the woody islands were cut every five years for supplying the riverine populations and the town of Lyon with fuel. On the margins of the floodplain, a few hardwood forests of *Fraxinus excelsior*, *Quercus robur* and *Acer pseudoplatanus*, escaped land clearances for pasture, subsistence agriculture and firewood cutting (Roux et al., 1989). More recently, Janssen et al. (2020) showed from aerial images that the critical changes undergone by the riparian forests of the Rhône River under persistent multiple stressors, strongly diverged from near-natural systems during the 20th century.

Historical maps and associated physical data have been used to assess fish-faunistic reference conditions in the framework of European biomonitoring improvement (Wolter et al., 2005; Ramos-Merchante et al., 2021). An historical analysis (from the end of the 19th century) of fish community structure in the Seine River catchment highlighted the major role of river continuity alteration in the decline of migratory species, the importance of introductions of non-native species, and changes in the proportion of rheophilic/eurytopic species in relation to the increase of human population and water pollution (Belliard et al., 2018). Historical data on fish communities along the Rhône River were discussed by Carrel (2002). In addition to fish data, Léger's maps available in the Arnaud et al. (2020a) dataset provide also morphological information on each river section (mean channel width, mean water depth). Combining physical data describing habitat heterogeneity (channel width, river bank length, diversity and accessibility of floodplain channels; see Hohensinner et al. 2004; 2011 for examples on the Danube River) at different periods with data on fish habitat preferences (Lamouroux et al., 1999; Rifflart et al., 2009) may be useful to achieve a better understanding of the influence of fluvial landscape transformation over the last few centuries on fish abundance and spatial patterns. More generally, the combination of biotic and abiotic archival data is important to assess how changes in hydromorphological river conditions influenced on biodiversity.

Finally, another potential use of the dataset is the reconstruction of extreme flood events based on historical channel geometries. The Beaucaire-Tarascon station (KP 269.5) records flows of the whole basin, and is the oldest hydrometric station on the Rhône. Daily water level measurements started in 1816, and daily estimates of discharge started in 1920 (Billy et al.,

2018). These chronicles are part of the HISTRHONE database, which contains a large volume of archives related to hydro-
climatic events in the lower Rhône over a period between the 14$^{th}$ and 20$^{th}$ centuries (Pichard and Roucaute, 2014). Further
work by hydrologists will involve exploiting historical topographic data prior to river training to reduce uncertainties in the
extrapolation of rating curves of hydraulic modelling based on original channel geometries. The topo-bathymetric data
highlighted in this study will be of particular interest (Fig. 6 and 8).

## 5 Data availability

The dataset is freely available in Pangaea at doi 10.1594/PANGAEA.922437 (Arnaud et al., 2020a). It contains the
iconographic database (12.7 GB), the .csv table describing the properties of the 350 data items (109 KB), the .csv table listing
the 281 archive box references (13 KB), and the GIS layer of the longitudinal coverage of the 350 data items (10.8 MB). The
dataset is available under the terms of the Creative Commons Attribution-NonCommercial (CC BY-NC) license, which permits
use, distribution and reproduction in any medium, provided that the use is not for commercial purposes and that the original
work is properly cited. This license was chosen to meet the data reuse policies of the 14 French archive services investigated
in this study, 4 of which have a reuse policy equivalent to the CC BY license (National Archives, AD Ain, AD Vaucluse,
DREAL, Univ. Grenoble), and 5 have a reuse policy equivalent to the CC BY-NC license (BnF, AD Rhône, AD Drôme, AD
Ardèche, AD Gard). Data from AD Isère, AD Savoie, CNR, and IGN, i.e. 10 items among the 350 items in the dataset, are not
furnished in the iconographic database. They are indicated as "available on request", because of specific reuse policies of these
institutions. The URLs of each institution's reuse policy are given in Pangaea metadata.

Extracts of the 350 data items are available in the public photo gallery of the Rhône Valley Human-Environment Observatory,
to enhance data visibility. This photo gallery uses the open source Piwigo software, with images being geolocated on an
interactive map. Map extracts have been oriented according to the flow direction of the Rhône, as long as this did not disturb
the toponym reading.

In addition to Pangaea, technical information on the dataset is available in the public metadata catalogue of the Rhône Valley
Human-Environment Observatory. Based on the open source GeoNetwork software, this metadata catalogue is harvested by
the LabEx DRIIHM (international consortium of Human-Environment Observatories) geocatalogue, which is then harvested
by the French geocatalogue and the European INSPIRE Geoportal, so that data discovery is possible at several hierarchical
levels (Arnaud et al., 2020b).

## 6 Conclusions

This study presents a historical database of 350 maps, topo-bathymetric cross-sections, and long profiles compiled on the
French Rhône River over the 1640–1950 period. Such an inventory of historical data over such a long time period and long
river course (530 km) has never been created before for the Rhône, and has rarely been created for other large rivers. This

database will enable further comparisons with other multi-impacted large rivers in Europe, such as the Danube, the Rhine, the

315 Ebro, and the Po, or even from one continent to another when data are available. The densification of archived time series is useful to detail and extend past trajectories of geomorphic, hydraulic, and ecological river features, and for conducting comparative analyses to detect differences and similarities between reaches and between rivers located in different geographical settings. Recently, rehabilitation measures have been extended to other by-passed reaches of the lower Rhône River (Baix-le-Logis-Neuf, Saint-Vallier, Montélimar), and these will also require analyses of channel changes over the last

two centuries to design restoration objectives. Some archive services were consulted in entirety for the time period and theme of this data article, notably the National Archives and AD Rhône, whereas additional investigations could be conducted in CNR or other departmental archives in coming years. Possible updates or extensions to the dataset might therefore happen in the future.

## Author contributions

All co-authors actively participated in the project "Collection and visualization of historical biophysical resources on the Rhône River" (OHM Vallée du Rhône, 2017-2019). Fanny Arnaud and Lalandy Sehen Chanu searched archive services, photographed historical resources, and prepared the database. Jules Grillot performed GIS treatment of scanned data. Jérémie Riquier brought his expertise on historical data compiled in previous long-term research on the Rhône River. Fanny Arnaud supervised the project and wrote the paper. Hervé Piégay, Dad Roux-Michollet, Jérémie Riquier, Georges Carrel, and Jean-

Michel Olivier revised the manuscript and provided important contributions in the section "Data use and application".

## Competing interests

The authors declare that they have no conflicts of interest in respect to this study.

## Acknowledgements

This research was funded by the OHM Vallée du Rhône in the LabEx DRIIHM French program "Investissements d'Avenir"

(ANR-11-LABX-0010), managed by the French National Research Agency (ANR). The work was performed in the EUR H2O'Lyon (ANR-17-EURE-0018) of Université de Lyon. Data were collected in Archives Nationales (France), Bibliothèque nationale de France, Archives départementales du Rhône et de la métropole de Lyon, Archives départementales de l'Ain, Archives départementales du Gard, Archives départementales de la Savoie, Archives départementales de l'Isère, Archives départementales du Vaucluse, Archives départementales de l'Ardèche, Compagnie Nationale du Rhône, Direction Régionale

de l'Environnement, de l'Aménagement et du Logement Auvergne-Rhône-Alpes, Institut National de l'information Géographique et forestière, Université de Grenoble. We thank colleagues from EVS involved in archive data collection since

2009: G. Fantino, P. Gaydou, E. Parrot, B. Räpple, N. Talaska. M. Lucas provided useful information on the Rhône hydrology. We thank N. Gastaldi and S. Rodriguez-Spolti from the Maps and Plans Service of National Archives for their help in the consultation of documents. K. Embleton proofread the manuscript.

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

**Table 1: Extract of the .csv table describing the properties of the 350 archive resources collected in this study. The symbol ~ in the Scale column means that it was not provided in the document, and the scale was therefore estimated by direct measurement of the distance between two benchmarks or by comparison with similar maps.**

**(a)**

| ID | Type | Source | Ref data | Original title (in French) | Translated title (in English) | Scale | Author(s) | Period |
|---|---|---|---|---|---|---|---|---|
| 44 | Map | AD01 | 100Fi 67 | Plan du cours du Rhône depuis Seyssel jusqu'à Pierre-Châtel sur le territoire de Virignin, ainsi que les communes avoisinantes | Map of the Rhône river from Seyssel to Pierre-Châtel on the territory of Virignin and neighbouring towns | Toises | *NA* | ~ 1750 |
| 264 | Map/Cross section | AN | CP/F/14/10074/1/A/piece25 | Projet de halage pour le passage du Pont-St-Esprit | Towing project for the Pont-St-Esprit passage | *~ 1:5000* | Grangent | 1812 |
| 261 | Map | AN | CP/F/14/10074/1/B/piece34 | Plan de la partie du cours du Rhône et de la rive gauche de ce fleuve comprise dans le département de la Drôme, entre l'embouchure du ruisseau du Dolon (...) et celle du Lauzon | Plan of the part of the Rhône river and the left bank of this river included in the department of the Drôme, between the Dolon confluence (...) and that of the Lauzon | 1:10 000 | Grailly | 1833 |
| 96 | Cross section | AN | F/14/6501 | Barrage du bras de Lavours | Dam of the Lavours channel | 1:1000 | O'Brien | 1835 |
| 255 | Map | AN | CP/F/14/10074/1/B/piece35 | Plan général de la rive droite du cours du Rhône depuis Limony, jusqu'au pont de Saint-Just. Troisième feuille de Meisse à St-Just | General map of the right bank of the Rhône river from Limony to the St-Just bridge. Third sheet from Meisse to Saint-Just | 1:10 000 | Ponts & Chaussées | 1835 |
| 304 | Map/Bathy-metry | AD69 | S 1434 | Amélioration du Rhône devant Beaucaire | Improvement of the Rhône river at Beaucaire | 1:10 000 | Goux | 1851 |
| 100 | Long profile | AD69 | S1356 | Nivellement en long sur les deux rives du Rhône | Longitudinal levelling on both banks of the Rhône river | 1:10 000 | Ponts & Chaussées | 1869 |
| 10 | Map | CNR, AD01 | *NA*, 100Fi 69 to 100Fi 114 | Plans Branciard | Branciard maps | 1:2000, 1:5000 | Branciard | 1910 |

**(b) Continuation of table 1.**

| Sector | Upstr. KP | Downstr. KP | Length (km) | Comment | Nb of images | Image acquisition | Storage file | Sub-file |
|---|---|---|---|---|---|---|---|---|
| Seyssel to Virignin | -151 | -117 | 34 | This map may have been surveyed after the Sardinian Maps of 1738 and before 1760 because 'lateral lines' from 1760 are not visible everywhere | 20 | Table | 1730-1875_Plans_UpperRhone | 1750_NA_Seyssel-Virignin |
| Pont-St-Esprit | 191 | 193 | 2 | Names of the bridge piers. 3 cross sections | 8 | Table | 1756-1822_Grangent_Plans | PontStEsprit_1812 |
| St Rambert to Pont-St-Esprit | 61.5 | 191.5 | 130 | Map of 12 m long. Name of some islands but not their nature. Left bank only. The Drôme River is visible | 25 | Table | 1833_Grailly_Plan_Dol-Lauz | |
| Lavours | -131.5 | | Ponctual data | 1 cross section | 8 | Table | 1799-1882_Topo-Bathy | 1835_OBrien_Lavours_CS |
| Meysse to St Just | 151 | 191 | 40 | Not the nature of bars and banks. Right bank only | 10 | Table | 1807-1858_PontsChaussees_Plans | Meysse-StJust_1835 |
| Beaucaire | 260 | 268.5 | 8.5 | 2 general maps, 3 detailed maps | 3, 6, 3, 3, 4 | Magnetic board | 1839-1853_Goux_Plans | Beaucaire_1851 |
| Seyssel to Lyon | -150 | 0 | 150 | Long profile indicating water depth and thalweg elevation | 14 | Table | 1869_PC_LP_Upper Rhone | |
| Virignin to Caderousse | -116.5 | 219 | 335.5 | Upstream of Lyon: AD01. Downstream of Lyon: CNR (available on request) | 47, 107 | HD scan, Georef 1st | 1910_Branciard_Plan_Viri-Cad | |

**Table 2: Georeferencing information on 14 map series of the database. The BD ORTHO® and PARCELLAIRE® base maps are provided by the IGN.**

| Translated title (in English) | Scale | Author(s) | Period | Sector | Length (km) | Nb img | Georeferencing transformation | Nb GCPs / img | RMS error (m) | Base map |
|---|---|---|---|---|---|---|---|---|---|---|
| Map of part of the Rhône river from Cordon to Hyenne (…) | 1:30 000 | *NA* | 18th c. | Yenne to Cordon | 25 | 2 | Spline | 12-24 | $10^{-5}$ | BD ORTHO® & PARCELLAIRE®, OpenStreetMap© |
| Geometrical map of part of the Rhône river from Geneva to the Guyer confluence (…) | 1:28 246 | De Bourcet | 1760 | Genève to St-Genix | 107 | 7 | 1st Order Polynomial & Spline | 13-25 & 20-26 | 34-44 & $10^{-5}$ | BD ORTHO® & PARCELLAIRE®, OpenStreetMap© |
| Map of the marshes and the plain from Beaucaire to Aiguemortes (…) | *~ 1:100 000* | Delisle | Late 18th c. | Beaucaire to the sea | 57 | 1 | Spline | 31 | $10^{-5}$ | BD ORTHO® & PARCELLAIRE®, OpenStreetMap© |
| Napoleonic land register | 1:10 000 | Napoleon I | 1810-1832 | St-Maurice to Piolenc | 63.5 | 16 | 1st Order Polynomial | *NA* | *NA* | BD ORTHO® & PARCELLAIRE® |
| The Rhône river: border between the Kingdom of France and the Duchy of Savoy | 1:14 400 | Corefa | 1831 | Cressin to St-Genix | 30 | 1 | 1st Order Polynomial & Spline | 23 & 38 | 59 & $10^{-5}$ | BD ORTHO® & PARCELLAIRE®, OpenStreetMap© |
| General plan and project for the dyke to be built against the highest floods of the Rhône (…) | 1:10 000 | Dignoscyo | 1837 | Jons to Pierre-Bénite | 31 | 2 | Spline | 50-85 | $10^{-5}$ | BD ORTHO® & PARCELLAIRE®, OpenStreetMap© |
| Map of the Rhône river between Donzère and Arles | 1:10 000 | Ponts & Chaussées | 1846 | Donzère to Arles | 116 | 12 | Spline | 19-52 | $10^{-5}$ | BD ORTHO® & PARCELLAIRE®, OpenStreetMap© |
| *NA* | 1:10 000 | *NA* | 1854 | Anglefort to Massignieux | 23 | 5 | Spline | 10-14 | $10^{-5}$ | BD ORTHO® & PARCELLAIRE®, OpenStreetMap© |
| Topographical map of the Rhône river | 1:10 000 | Ponts & Chaussées | 1857-1876 | Surjoux to the sea | 489 | 58 | 1st Order Polynomial | ~ 15. Improvement of 9 images: 10-17 | < 50. Improvement of 9 images: 4.6-12.8 | BD ORTHO® |
| Completion and first maintenance works | 1:10 000 | Girardon | 1891 | Lyon to Donzère | 169.5 | 55 | 1st Order Polynomial | 6-15 | 3.0-9.5 | BD ORTHO® |
| Bathymetric maps of the Rhône river | 1:10 000 | Ponts & Chaussées | 1897-1908 | Lyon to the sea | 307 | 66 | 1st Order Polynomial | 4-12 | 0.6-20.5 | BD ORTHO® |
| Branciard maps | 1:2000, 1:5000 | Branciard | 1910 | Virignin to Caderousse | 170 | 107 | 1st Order Polynomial | 3-15 | 0.3-11.3 | BD ORTHO® & PARCELLAIRE® |
| General staff maps | 1:20 000-1:80 000 | Etat-Major | 1832-1941 | Genève to the sea | 535 | 162 | 1st Order Polynomial | 4-6 | 0.8-26.3 | BD ORTHO® |
| Pre-work plans | 1:10 000 | CNR | 1946-1983 | Anglefort to Caderousse | 162 | 128 | 1st Order Polynomial | 4 | *NA* | BD ORTHO® |

List of Figures' captions

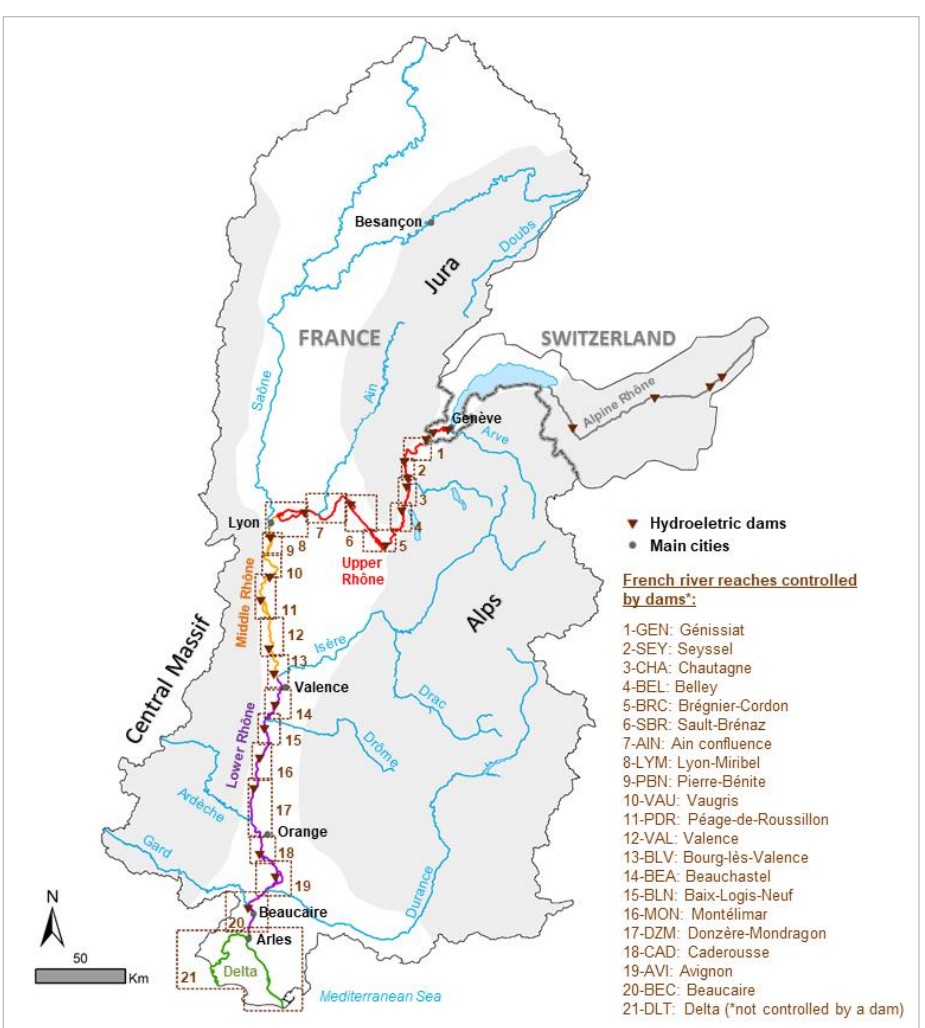

Figure 1: The Rhône River corridor in the French-Swiss basin.

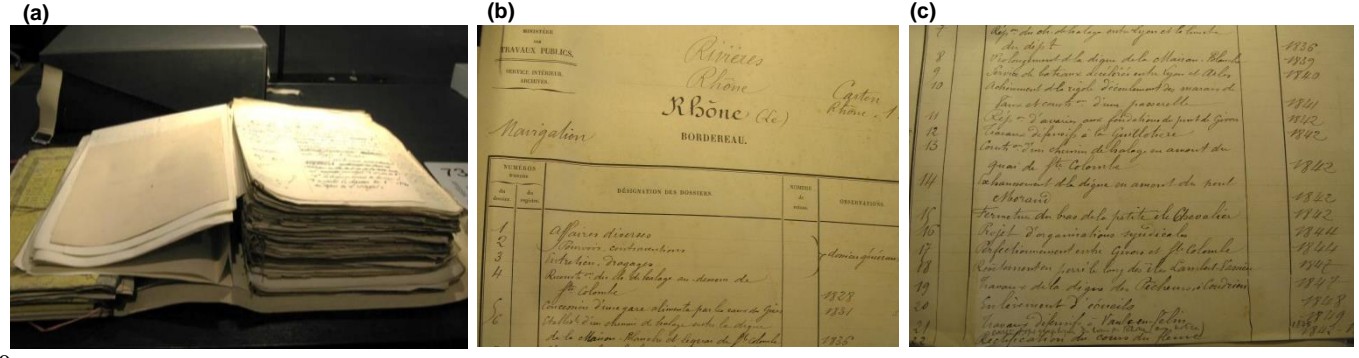

**Figure 2: Example of the box F/14/6706 in the National Archives (online catalogue information: 'Rhône River, Files 1-22, 1828-1849'): (a) bundle of documents; (b) and (c) content description.**

**Figure 3: Examples of the archive resources collected in this study: (a) map series at 1:28 246 between Geneva and the Guiers confluence, 1760. Extract of sheet n°3/7 (source: De Bourcet, AD Savoie, 1Fi S52-53); (b) bathymetric information on a map at 1:2000 in Lyon, 1833 (source: Garella, AD Rhône, S 1361); (c) long profile of the left bank and the thalweg of the upper Rhône, 1869. Extract of the sheet n°6/6 (source: *Ponts et Chaussées*, AD Rhône, S 1356); (d) map series at 1:10 000 from Geneva to the Mediterranean Sea, 1842. Extract of the sheet n°20/25 (source: Arnaud, National Archives, CP/F/14/10074/2/Piece 8).**

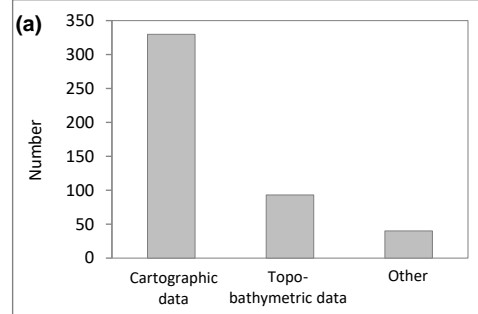
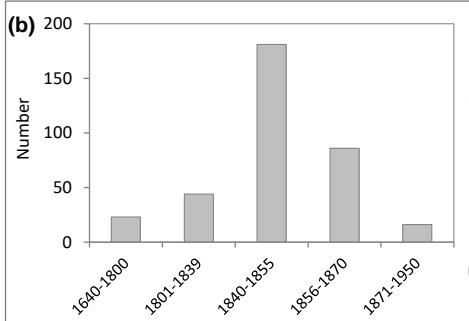

**Figure 4: (a) Number of archive data items compiled in this study. "Other" corresponds to plans <1:1000, long profiles of the Swiss Rhône, and books. (b) Temporal extent of cartographic and topo-bathymetric data.**

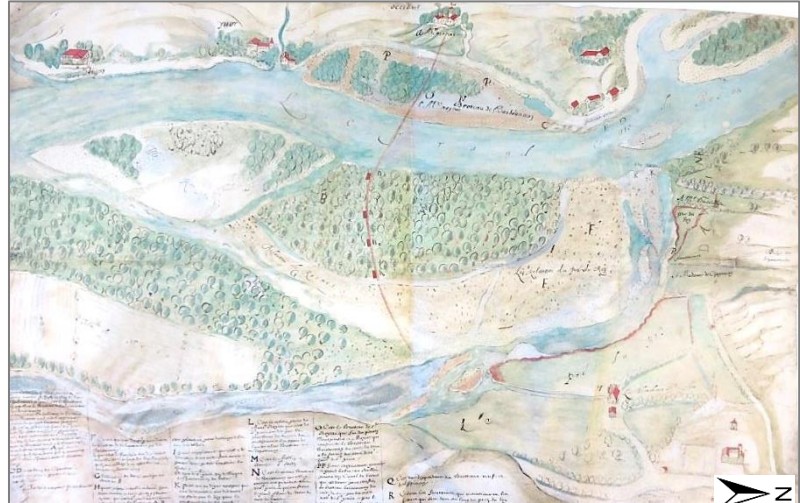

**Figure 5: Map at ~1:10 000 between Pierre-Bénite and Irigny, 1644 (source: Maupin, AD Rhône, 44 J 336).**

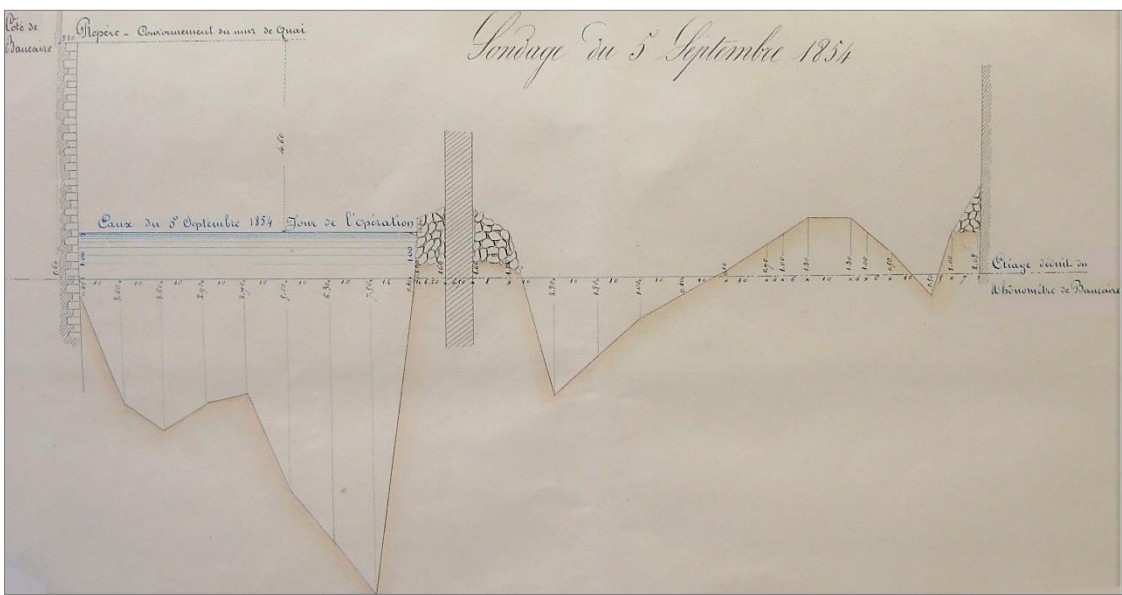

**Figure 6: Cross-section at the suspension bridge of Beaucaire-Tarascon. The low-flow water level deducted from the Beaucaire gauging station and the water level on the day of the survey (September 5, 1854) are indicated in blue (source: Aymard, AD Rhône, S 1434).**

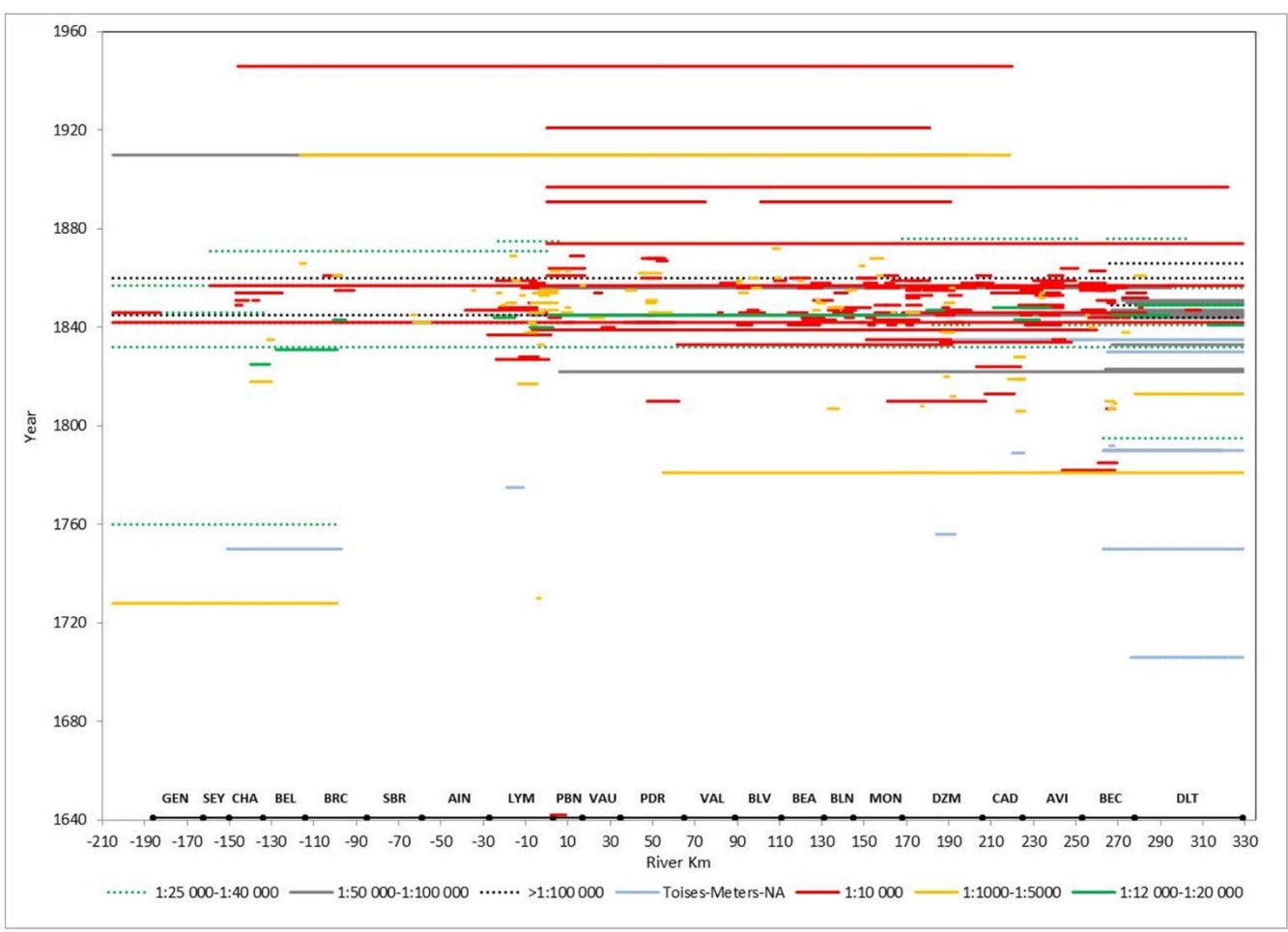

**Figure 7: Longitudinal coverage and spatial scales of the cartographic dataset. The spatial extent of present river reaches is superimposed on the x-axis (see Fig. 1 for acronyms).**

610

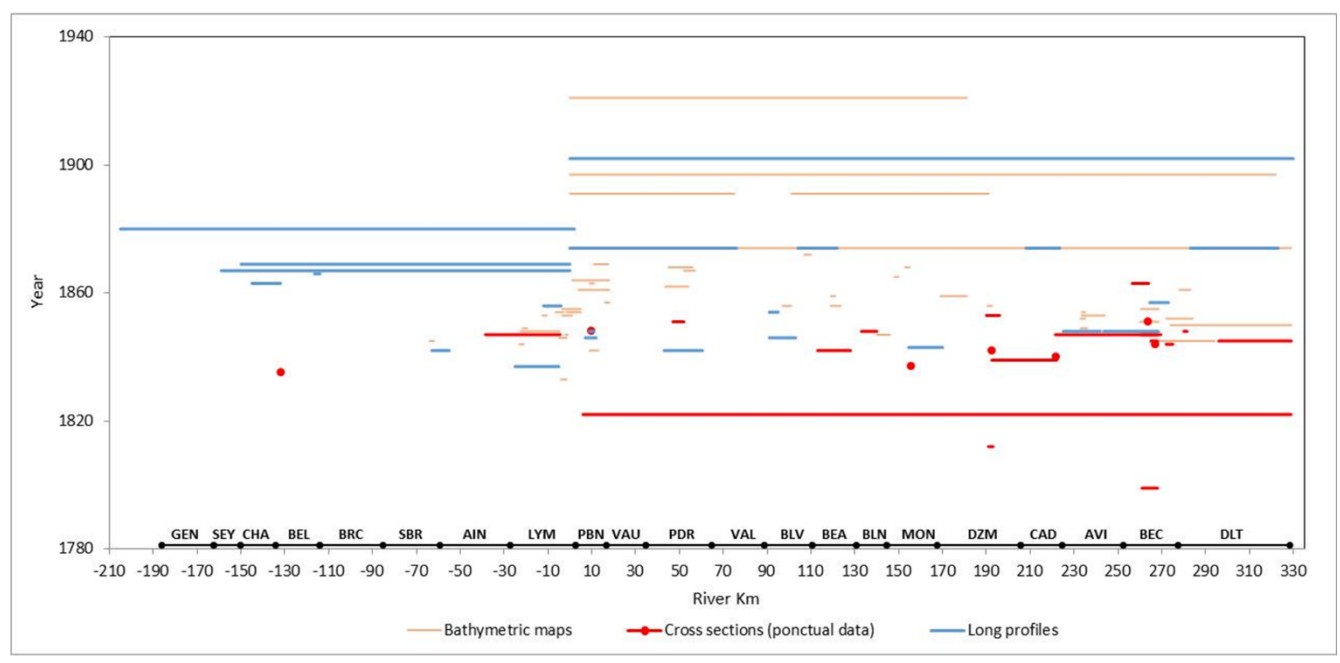

**Figure 8: Longitudinal coverage of the topo-bathymetric dataset. The spatial extent of present river reaches is superimposed on the x-axis (see Fig. 1 for acronyms).**

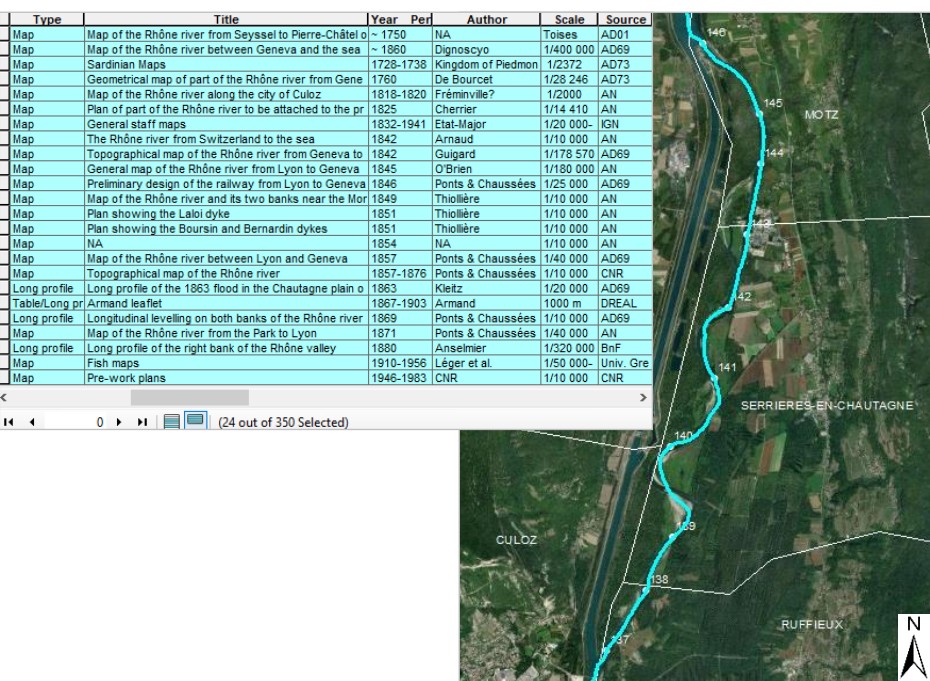

**Figure 9: GIS layer of the longitudinal coverage of data collected in this study. Example of the 24 available data items on the Chautagne river reach (KP −146 to −137). Basemap: Esri.**

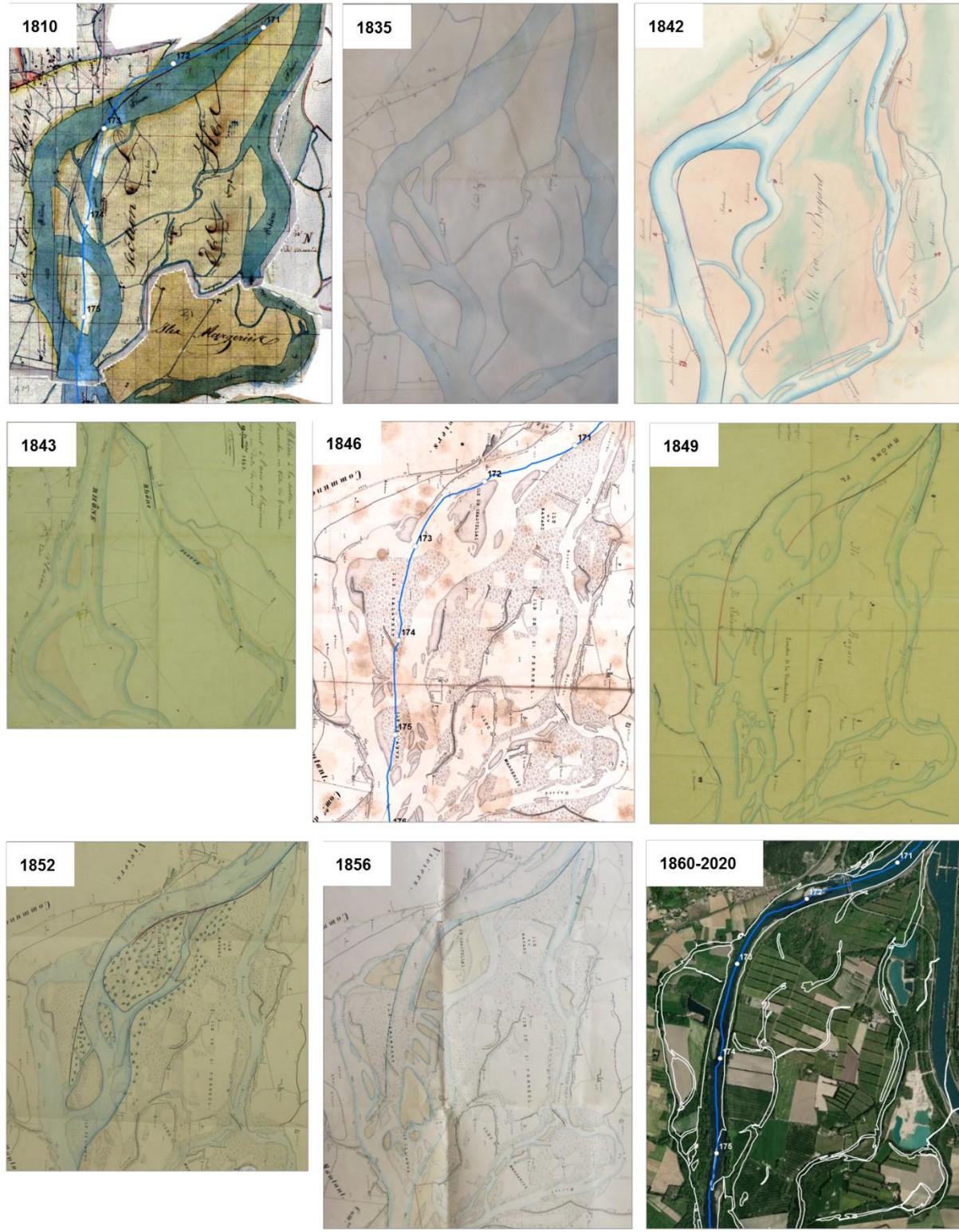

635

**Figure 10: Examples of maps surveyed on the upstream section of the Donzère-Mondragon reach, 1810–1860. The present Rhône centreline is superimposed on the 1810, 1846 and "1860-2020" georeferenced maps. The scale is given by the spacing between the Kilometric Points (KPs) on these maps. The "1860-2020" map shows the present orthophotograph, on which the delineations of aquatic channels in 1860 were superimposed (Source: 1810: Napoleonic Land Register; 1835, 1846, 1856 and 1860: *Ponts et Chaussées*; 1842: Arnaud; 1843: Bouvier; 1849 and 1852: Goux; 2020: Esri).**

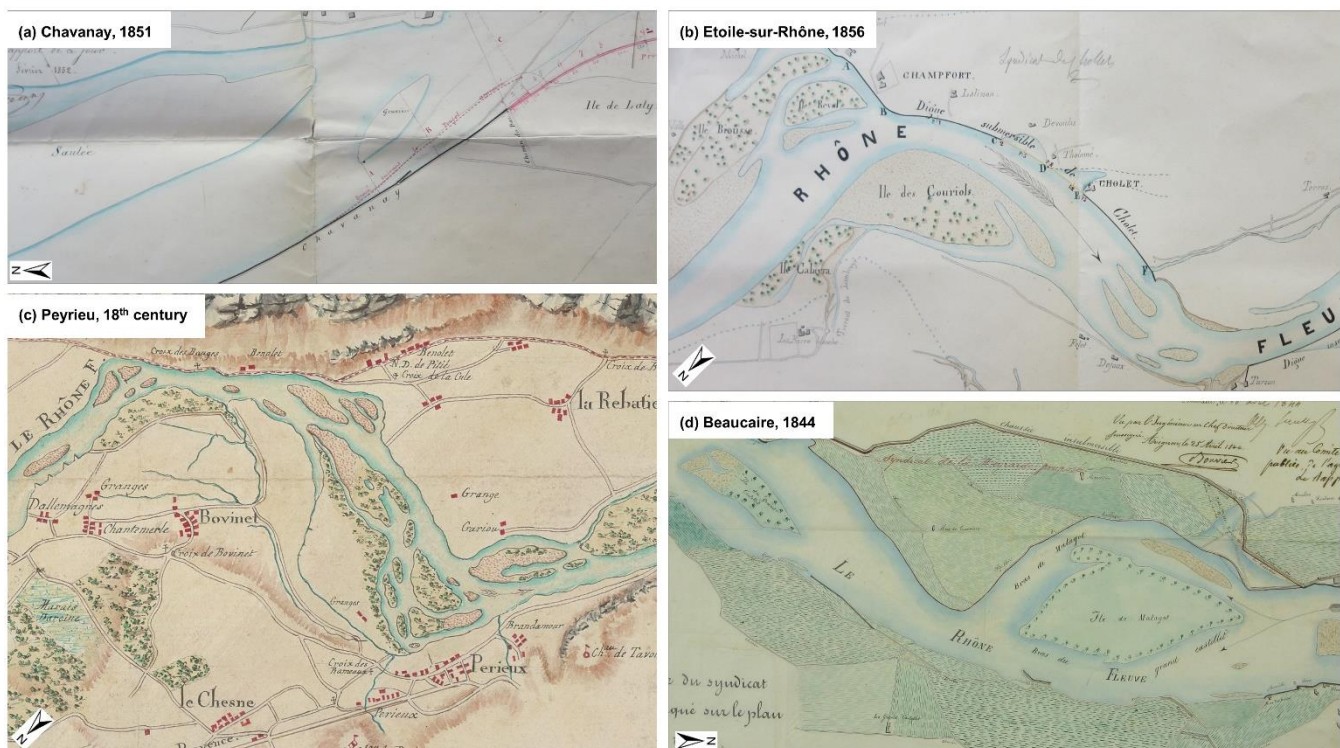

**Figure 11: Examples of vegetation representation, from the simplest to the most detailed: (a) island annotation: 'saulée' in French means 'willows', 'gravier' means 'gravel' (source: Goux, 1851, AD Rhône, S 1393); (b) riparian vegetation close to the channel (source: Kleitz, 1856, AD Rhône, S 1397); (c) depiction of vegetation in the floodplain (source: unknown author, 18th century, National Archives, CP/F14/10074/1/A/Piece3); (d) differentiation between vegetation types (source: Bouvier, 1844, National Archives, F/14/6583).**

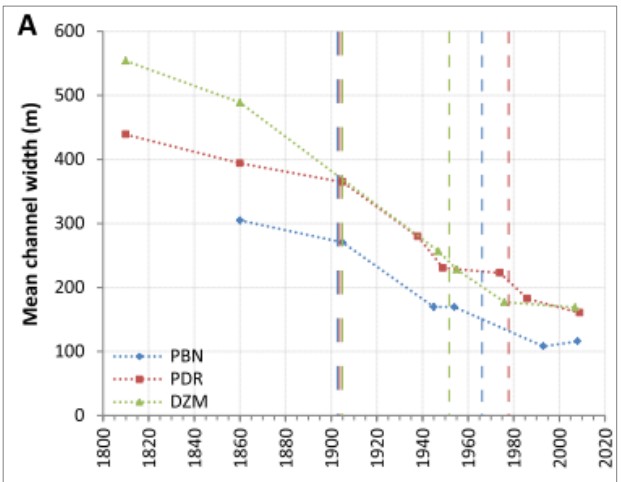

**Figure 12: Temporal variation in the mean active channel width of three reaches of the Rhône between 1810 and 2009. Long dashed lines correspond to the Girardon training phase and short dashed lines correspond to the channel by-passing for each reach (Tena et al., 2020).**