# Peer review of "Historical cartographic and topo-bathymetric database on the French Rhône River (17th–20th centuries)"

_Earth System Science Data, 2020_

## Referee Comment (RC1) · Anonymous Referee #1 · 8 Jan 2021

It is very important in fluvial studies to collect as long dataset as possible, because it is the only one way to understand the evolution of rivers. Within this study large amount of data were collected and made available, providing a good database for further studies. I expected throughout of the text (and the tables) to have some kind of scientific note on the accuracy of the maps and surveys, which are important for a user: if they want to understand the changes: e.g. the map was made after a large flood, or after a long-lasting dry period, or at low stage etc. Anything, what could help the work of future users. . .. You referred several times to Arnaud et al. 2020. But it is not clear for the reader, why/how is this article more or different compared to that publication. Please, clear it! L. 65. Please, reformulate these sentences, and indicate what are the aims of

your research. L. 70: Study area: Would be great to have some data on the regime of the river, like annual timing of low stages and floods, duration of floods, differences between the lowest and highest stages, and how did these parameters change. It is important, as the hydrology of the river influences the result of the mapping, thus the precision of the derived data. L. 110-115: It is not necessary to mention, that how many days you had spent in Archives. Usually a research takes days and months. . . Fig 1. What is indicated by the grey colour? Please add to the legend! Fig.9. Please provide a scalebar and the north direction (as on Fig. 3 and 5 you use various directions) Table1. I do not understand this table, as it has two very distinct parts. Would better to split it into two parts.

---

## Referee Comment (RC2) · Hanna Hajdukiewicz (Referee) · 11 Jan 2021

General comments: The manuscript provides an inventory of very extensive historical database on the French Rhône Rive from the 17th to mid-20th century. The presented dataset is very imposing and include: numerous historical maps, plans, cross-sections, topo-bathymetry data items and profiles. Such a dataset is particularly useful for proper reconstruction of the river channel evolution and past fluvial processes and can be especially applied in river revitalization planning process. On the basis of this data the changes of the river hydromorphological condtition can be assessed.

Specific comments: The data items was collected on one layer in ArcMap 10.5, which

facilitated the quick use of dataset.

The authors tested the spline transformation on archival maps. This isue requires more clarification why the second or third order polynomial transformation wasn't applied.

The article is based on rich references to literature. In my opinion the dataset is complete, additionally the informations about 350 archive resources and about georeferencing on maps are depicted clearly in the tables. For each map the RMS terror is estimated. The archival maps are geo-referenced and can be used to compare river channel elements between time horizons in ArcGIS and QGIS softwares.

I suggest to consider adding the chapter 5: 5. Limitations resulting from inaccuracies of archival materials (or critical analysis of the dataset). In this chapter authors can collect all the limitations of archival maps that make it impossible to analyse some long-term environmental changes (for example, exact area size changes or the type of forest changes). It should contain the considerations if the use of such materials (maps from the 18th and 19th centuries) for the analysis of smaller rivers of Europe give good results or is it sufficient only for large rivers?

Figures and tables are correct and high quality. The information in the figure 6 and 7 are especially valuable allowing to quickly find out about the availability, scale, time horizon and topo-bathymetric information for given river reach. I suggest only to enlarge the figure 11 to make the letters and numbers more readable.

The dataset publication is of high quality. Collecting data on a layer in ArcGis will enable their easy use in the future for various studies.

Detailed comments are icluded in PDF.

Please also note the supplement to this comment:
https://essd.copernicus.org/preprints/essd-2020-274/essd-2020-274-RC2-supplement.pdf

[Figure]

**Supplement:**

[revised manuscript text omitted]

---

## Author Comment (AC1) · 23 Mar 2021

**Anonymous Referee #1 - Received and published: 8 January 2021**

It is very important in fluvial studies to collect as long dataset as possible, because it is the only one way to understand the evolution of rivers. Within this study large amount of data were collected and made available, providing a good database for further studies.

**Thank you for your positive comment.**

I expected throughout of the text (and the tables) to have some kind of scientific note on the accuracy of the maps and surveys, which are important for a user: if they want to understand the changes: e.g. the map was made after a large flood, or after a long-lasting dry period, or at low stage etc. Anything, what could help the work of future users. . ..

We added hydrological information on maps and topo-bathymetric data lines 195-200 and we cited map objectives lines 240-243. We replaced Figure 11 by Figure 6 which better illustrates hydrological annotations found on a cross-section.

You referred several times to Arnaud et al. 2020. But it is not clear for the reader, why/how is this article more or different compared to that publication. Please, clear it!

Arnaud et al. 2020a is the dataset deposited in Pangaea. According to ESSD guidelines, we cited the dataset several times in the manuscript. We clarified the citation.

L. 65. Please, reformulate these sentences, and indicate what are the aims of your research.

**Done.**

L. 70: Study area: Would be great to have some data on the regime of the river, like annual timing of low stages and floods, duration of floods, differences between the lowest and highest stages, and how did these parameters change. It is important, as the hydrology of the river influences the result of the mapping, thus the precision of the derived data.

See our previous response on the addition of hydrological information on the dataset. We also added a paragraph on the flow regime of the Rhône River for more than 300 years, lines 88-100.

L. 110-115: It is not necessary to mention, that how many days you had spent in Archives. Usually a research takes days and months. . .

We deleted the number of days.

Fig 1. What is indicated by the grey color? Please add to the legend!

The grey color indicates the mountain massifs. We completed the legend.

Fig.9. Please provide a scalebar and the north direction (as on Fig. 3 and 5 you use various directions)

Done for the North direction. The scale is given by the spacing between the Kilometric Points (KPs) on the three georeferenced maps. The other maps were manually adjusted to cover the same area. We specified it in the caption.

Table1. I do not understand this table, as it has two very distinct parts. Would better to split it into two parts.

The second part is the continuation of table 1 with the same eight archive resources taken as examples. We specified it in the caption.

Hanna Hajdukiewicz (Referee) hajdukiewicz@iop.krakow.pl - Received and published: 11 January 2021

General comments: The manuscript provides an inventory of very extensive historical database on the French Rhône Rive from the 17th to mid-20th century. The presented dataset is very imposing and include: numerous historical maps, plans, cross-sections, topobathymetry data items and profiles. Such a dataset is particularly useful for proper reconstruction of the river channel evolution and past fluvial processes and can be especially applied in river revitalization planning process. On the basis of this data the changes of the river hydromorphological condition can be assessed.

**Thank you very for your positive feedback on the manuscript.**

Specific comments: The data items was collected on one layer in ArcMap 10.5, which facilitated the quick use of dataset. The authors tested the spline transformation on archival maps. This issue requires more clarification why the second or third order polynomial transformation wasn't applied.

Half of the 14 scanned map series were collected and georeferenced several years ago. Practices in our lab were to apply a first order polynomial transformation to old maps and to reserve the second and third orders to aerial photographs with mountainous reliefs. For this study, we based on recommendations of Lestel et al. 2018 who georeferenced dozens of maps using the Spline transformation. Before applying this, we also tested the 2nd and 3rd orders but the positioning was always better when using Spline. We changed the text accordingly.

The article is based on rich references to literature. In my opinion the dataset is complete, additionally the information about 350 archive resources and about georeferencing on maps are depicted clearly in the tables. For each map the RMS terror is estimated. The archival maps are geo-referenced and can be used to compare river channel elements between time horizons in ArcGIS and QGIS softwares.

**Thank you again for your positive comments.**

I suggest to consider adding the chapter 5: 5. Limitations resulting from inaccuracies of archival materials (or critical analysis of the dataset). In this chapter authors can collect all the limitations of archival maps that make it impossible to analyze some long-term environmental changes (for example, exact area size changes or the type of forest changes).

We completed Section 4 following your comment. See lines 251-256 for the critical analysis of the dataset regarding vegetation study and lines 288-290 for fish study.

It should contain the considerations if the use of such materials (maps from the 18th and 19th centuries) for the analysis of smaller rivers of Europe give good results or is it sufficient only for large rivers?

We refer to « large rivers » because the Rhône is one of the largest rivers in Europe and is often compared to the Danube and the Rhine as similar large medio-European river systems. Such archive materials could be used to analyze smaller rivers, depending on data availability and map scale. See e.g. works by Lestel et al. 2018 who compiled old maps on the entire Seine river watershed, including small tributaries; Scorpio et al. 2015 on medium-size rivers of Southern Italy; Hohensinner et al. 2021 on alpine river catchments from 500 to 13 400 km2. Note also that some maps of our dataset depict Rhône River basin tributaries (see historical fish maps from Léger et al.) or tributary confluences, i.e. small and medium-size rivers. We reduced the number of "large" rivers citations in the manuscript. We cited Scorpio et al. (2015) and Hohensinner et al. (2021) line 42 and we cited tributary confluences line 259.

Figures and tables are correct and high quality. The information in the figure 6 and 7 are especially valuable allowing to quickly find out about the availability, scale, time horizon and

topo-bathymetric information for given river reach. I suggest only to enlarge the figure 11 to make the letters and numbers more readable. The dataset publication is of high quality. Collecting data on a layer in ArcGis will enable their easy use in the future for various studies.

We replaced Figure 11 by Figure 6 to respond to Reviewer 1.

**Detailed comments are included in PDF:**

L.45: Could you consider to add there also the sentence about river hydromorphological condition changes assessment on the basis of the set of archival materials?

**Done line 40.**

L.57: Could you explain shortly why? Whether the shape of the river channel (braided and anabranching) in the mid-19th century was the result also of human interference like a massive deforestation. How did the river look like in the 18th century, was the river valley more forested and is it depicted on the maps from the 18th century?

Human manipulations of the river were still relatively modest in the mid-19th century (Bravard, 2010). Extensive river training, with the systematic construction of "unsubmersible" dykes, began at the end of the 19th century. Diachronic studies used the map series of 1860 because it is a very detailed representation of the Rhône channel and its floodplain. We completed lines 59-62. The Rhône was a braided and anabranching river in the 18th century (line 55). Roux et al. (1989) mentioned the agricultural landscape of the upper Rhône in this period, with woody islands regularly cut and a few hardwood forests on the margins of the floodplain. We completed lines 267-271. Diachronic analysis of vegetation could be conducted by taking care of vegetation representation on maps (see other responses on vegetation analysis).

**L.61: not listed in references**

**We added Thorel et al. 2018 in the reference list.**

L.191: Please explain more why you didn't applied the second or third polynomial transformation for maps from 1760 and 1831?

See our response to the first specific comment. We changed the text accordingly.

L.195: Could you estimate which cartographic materials (from which dates) are or could be georeferenced with RMS error not exceeding 5% of the total river width (are sufficiently cartometric) that they would be suitable for calculations the parameters of active river zone that you mention in this sentence, like channel width, gravel bar areas...For example if every maps from 19th century could be used for this or there is some "border date" (1860?).

Among the 330 maps of the database, 14 were georeferenced (see section 3.1). It is difficult to give a % estimate of georeferencing based on RMS error because in the case of Spline transformation, the resulting RMS error is not expressed in meters. Most of the other maps of the database can be georeferenced by using riverine roads, buildings and dykes as GCPs, even maps before 1860 (see Table 2). Georeferencing depends mainly on the map scale and the depicted features. We specified these details lines 221-223.

L.232: Could you add information about possibilities of performing long-term forest changes analyses on the basis of maps symbols. Whether the symbols are the same on the maps from different time horizons?

We added considerations on vegetation mapping (either with annotations, colors or symbols) and we added Figure 11. See line 251-256.

L.243: This is very important issue that the application of the archival biotic and abiotic environments data indicates how changes in hydromorphological river conditions influenced on biodiversity. I suggest to add this information in one or two sentences.

Thank you for your suggestion. We improved the text lines 294-296.

L.455: This is the part of the table 1? In my opinion there should be information how to read this table like "continuation of table 1".

You are right. See the answer given to reviewer 1 (his last comment).